

# Precipitation extremes in a EURO-CORDEX 0.11° ensemble at hourly resolution

Peter Berg[1], Ole B. Christensen[2], Katharina Klehmet[1], Geert Lenderink[3], Jonas Olsson[1],
Claas Teichmann[4], and Wei Yang[1]

[1]Swedish Meteorological and Hydrological Institute, Folkborgsvägen 17, 610 76 Norrköping, Sweden
[2]Danish Meteorological Institute, Lyngbyvej 100, 2100 Copenhagen, Denmark
[3]KNMI Royal Netherlands Meteorological Institute, Utrechtseweg 297, 3731 GA De Bilt, the Netherlands
[4]Climate Service Center Germany (GERICS), Helmholtz-Zentrum Geesthacht, Fischertwiete 1, 20095 Hamburg, Germany

**Correspondence:** Peter Berg (peter.berg@smhi.se)

**Abstract.** Regional climate model simulations have routinely been applied to assess changes in precipitation extremes at daily
time steps. However, shorter sub-daily extremes have not received as much attention. This is likely because of the limited
availability of high temporal resolution data, both for observations and for model outputs. Here, summertime depth duration
frequencies of a sub-set of the EURO-CORDEX 0.11° ensemble is evaluated with observations for several European countries
for durations of one to 12 h. Most of the model simulations strongly underestimate 10-year depths for durations up to a few
hours, but do better on 12 h durations. All models fail in reproducing observed spatial patterns over Germany for durations
shorter than 12 h, but all reproduce the pattern at least partly at 12 h duration. Large-scale driven spatial patterns, such as
the extreme depths in southern France are better captured also at shorter durations, albeit severely underestimated. Projected
changes are assessed by relating relative depth changes to mean temperature changes. A strong relationship with temperature
is found across sub-regions of Europe, the emission scenario and future time period. However, there is an equally strong
dependency on the global and regional model applied, with a spread in scaling of around 1–10%/K at 12 h duration, and
generally higher values at shorter durations.
# 1  Introduction
Cloudbursts are the result of enormous quantities of atmospheric water vapour being concentrated to a relatively small area for
a short duration. The natural and societal landscape has large problems to cope with the huge amounts of water, with resulting
issues of local flooding, damages to infrastructure, landslides, erosion, etc. Theory predicts an intensification of cloudbursts
with a warming climate (Trenberth et al., 2003), which makes modelling of future projections important to aid planning
of robust infrastructure as well as methods to cope with diversion or delays of water in especially urban settings. Global
climate models (GCMs) are generally of too coarse spatio-temporal resolution to allow detailed analysis, but some state-of-
the-art regional climate model (RCM) ensemble members provide precipitation output at sufficient resolution for analysis of
cloudburst statistics.





Short duration extremes are often studied from an urban planning perspective, where the consequences of insufficient in-
frastructure to deal with cloudbursts can be catastrophic (Willems et al., 2012). A common approach for cloudburst analysis
is to investigate mean intensities, or depths, as a function of duration and to perform extreme value analysis to determine
depth-duration-frequency (DDF) functions. Mid-latitude cloudburst have a typical dimension of 10–100 km and a duration of
one to several hours, which sets the scale of any record for studying these type of events. For example, the highest recorded
cloudburst in Sweden (in gauge observations between 1996 and 2017) lasted in total for 3 h, but with extreme intensities of
about 17 and 40 mm/15min for only two consecutive measurements. Still, the event holds the record for durations up to a few
hours.
The EURO-CORDEX ensemble of high resolution, 0.11° (about 12 km), simulations provide the first larger ensemble with
sufficient spatial resolution for studying cloudbursts (Kotlarski et al., 2014). However, RCMs and GCMs have shown severe
problems with their sub-grid scale parametrisations of convective processes, which affect their ability to reproduce, e.g., the
diurnal cycle of rainfall intensity (Trenberth et al., 2003; Fosser et al., 2015; Prein et al., 2015), the peak storm intensities
(Kendon et al., 2014), and extreme hourly intensities (Hanel and Buishand, 2010). It is therefore questionable to which extent
such RCMs are capable of describing cloudbursts in present as well as in future climate.
Olsson et al. (2015) presented increasing agreement of modelled and observed hourly precipitation with higher spatial
resolution, and 6 km resolution of a parametrised RCM (RCA3) is in approximate agreement with gauge observations. Similar
results were obtained for Denmark, where also future projections were found to show larger increases in extreme precipitation
for higher spatial resolution and shorter temporal aggregations (Sunyer et al., 2016). Convective permitting regional models at
less than about 5 km resolution, have been shown to better simulate the peak structure of extreme events (Kendon et al., 2014),
better agreement with observations regarding the diurnal cycle of precipitation intensity (Fosser et al., 2015; Prein et al., 2015),
as well as improved performance of extreme hourly events (Ban et al., 2018).
The fate of cloudbursts in a warming climate is tied to the availability of atmospheric water vapour. A warmer atmosphere can
hold more water, following the Clausius-Clapeyron (CC) equation. At average mid-latitude conditions, the moisture holding
capacity of the atmosphere increases at a rate of about 7%/K (CC-rate), and e.g. Trenberth et al. (2003) argue that extreme
convective precipitation, i.e. cloudbursts, are expected to intensify at or even beyond the CC-rate in a warming climate. Studies
of the scaling of cloudbursts with temperature from present-day day-to-day variability has shown increases beyond the CC-
rate (e.g. Lenderink and van Meijgaard, 2008; Berg et al., 2013; Westra et al., 2014). How such studies relate to changes
in climate is debated (Bao et al., 2017; Barbero et al., 2018), and also trend analysis of cloudbursts suffer from short and
non-homogeneous records leaving any potential trends unclear or non-significant (Willems et al., 2012). There are, however,
some studies of precipitation extremes that present observational support for the super CC-rate derived from long term trends
in a warming climate (Guerreiro et al., 2018; Westra et al., 2013). Further, data from GCM and RCM data are generally of
too coarse spatio-temporal resolution for detailed evaluation of their performance and analysis of their future projections. The
scaling of hourly precipitation with increasing temperature in future projections has generally been shown to be constrained to
the CC-rate, but sometimes also stronger scaling is seen with higher resolution convection permitting models (Kendon et al.,
2014; Fosser et al., 2017; Ban et al., 2015, 2018). While these high resolution simulations show increased performance, their





**Table 1.** The RCM-GCM simulations with hourly precipitation output that are included in the analysis. The experiment code ("rip-nomenclature") from CMIP5 indicates the realization (r), the initialization (i) and the physics set-up (p) used. Here, the code is listed due to differences in the realizations of the EC-Earth model.

| Name | RCM | GCM | Experiment | Institute |
|------|-----|-----|-----------|-----------|
| RCA4-EC-Earthr12 | RCA4 | EC-Earth | r12i1p1 | SMHI |
| RCA4-CNRM-CM5 | RCA4 | CNRM-CM5 | r1i1p1 | SMHI |
| RCA4-MPI-ESM-LR | RCA4 | MPI-ESM-LR | r1i1p1 | SMHI |
| RCA4-IPSL-CM5A-MR | RCA4 | IPSL-CM5A-MR | r1i1p1 | SMHI |
| RCA4-HadGEM2-ES | RCA4 | HadGEM2-ES | r1i1p1 | SMHI |
| RACMO22E-HadGEM2-ES | RACMO22E* | HadGEM2-ES | r1i1p1 | KNMI |
| RACMO22E-EC-Earthr01 | RACMO22E | EC-Earth | r1i1p1 | KNMI |
| HIRHAM5-EC-Earthr03 | HIRHAM5 | EC-Earth | r3i1p1 | DMI |
| REMO2009-MPI-ESM-LR | REMO2009 | MPI-ESM-LR | r1i1p1 | GERICS |

* Version 2 (v2) of the simulation as submitted to the Earth System Grid Federation (ESGF).

availability is still limited. Therefore, the current state-of-the-art regional climate model ensemble that is being applied for climate services and local assessments for adaptation is the EURO-CORDEX 0.11° ensemble, which we explore here.

In this study, we evaluate the performance of four state-of-the-art regional climate models with hourly output frequency, in their ability to reproduce observed DDF statistics across Europe for the summer half-year. Future projections under the RCP4.5 and RCP8.5 emission scenarios are then investigated, and the scaling of extreme precipitation statistics with temperature is explored. The paper starts with a presentation of the data sources (Section 2), followed by the applied methodology (Section 3), results of the evaluation and future projections (Section 4), and ends with a discussion (Section 5) and the main conclusions (Section 6).

## 2 Data

### 2.1 The EURO-CORDEX ensemble

EURO-CORDEX at 0.11° spatial resolution is the current state-of-the-art regional climate model ensemble over Europe. The ensemble is the result of the cooperation between many European institutions and further ensemble members are still being added. Here, we are limited to a sub-set of the ensemble with members for which we have received precipitation data at one hour temporal resolution, see Table 1. This sub-set is not including the common reanalysis downscaling simulations, and the analysis is therefore of GCM-RCM combinations which introduces some additional uncertainties (Déqué et al., 2012).

Kotlarski et al. (2014) gives an overview of the details of the models and applied parametrisations, and also presents the performance of the RCMs in reanalysis driven simulations, mainly discussing average quantities of precipitation and temperature. Focusing on their results for the summer season, the RCMs in the sub-ensemble used here follow the general pattern of a warm



summer bias in REMO2009 in continental Europe, whereas RACMO22E has a general cold bias, and RCA4 and HIRHAM5 are too warm in the south and too cold in the north. Bias in precipitation is more scattered, but follows a similar structure as the temperature bias for each of the models, indicating a strong dependency of cold and wet conditions, as can be expected for mean quantities. Prein et al. (2016) show that model bias in the EURO-CORDEX 0.11° simulations are reduced compared to the earlier 0.44° simulations, for both mean and extreme daily and 3-hourly precipitation, especially in local areas. Rajczak and Schär (2017) analysed heavy and extreme daily precipitation intensity and found good performance in RCMs, mostly independent of the driving GCM.

Jacob et al. (2014) investigated end-of-century climate change for the EURO-CORDEX 0.11° simulations, with significant changes in both mean precipitation and temperature across Europe for RCP4.5 and 8.5. Whereas mean precipitation generally increases in Northern Europe and decreases in Southern Europe, heavy precipitation shows robust changes across the ensemble, with significant increases in north-eastern Europe in summer, and pan-European increases in winter under RCP8.5. Kjellström et al. (2018) investigated climate change patterns as a function of global mean temperature increases of 1.5 and 2.0°C, with similar results for mean precipitation and temperature as in Jacob et al. (2014). Projected precipitation extremes were investigated by Dosio (2015), and showed general increases in the annual top daily extreme and in the 95th percentile of the precipitation distribution.

The presented analysis makes use of a historical period from 1971–2000, as well as future scenario periods 2011–2040, 2041–2070, and 2071-2100. The analysis is restricted to summer-half years (April–September) to focus on the main convective season in Europe (Berg et al., 2009). Representative concentration pathways (RCP) 4.5 and 8.5 are investigated for all models.

## 2.2 National DDF data

The model simulations are evaluated against gauge based DDF curves as obtained from countries across Europe, namely Austria, Germany, Sweden, the Netherlands, and France. Much of the information about how the DDFs were calculated is only available in local language, and the exact procedures are sometimes not clearly or sufficiently explained. Below, we provide a brief introduction to each data set, but refer to references for details.

### 2.2.1 Sweden

The Swedish DDFs statistics were recently updated by Olsson et al. (2018a). The statistics are based on about 125 gauge observations, with a fixed 15-min measurement interval, and with data for the period 1996–2017. Durations of 15 min to 12 h were studied, using the block rainfall method, and corrected for underestimations due to the fix 15 min interval by multiplication by 1.18, 1.08, 1.041, 1.036, and 1.029 for durations 15 min, 30 min, 45 min, 1 h, and 2 h, respectively. No correction was deemed necessary for longer durations. The coefficients were derived by comparison with additional tipping-bucket gauges, and agrees approximately with earlier studies (Malitz and Ertel, 2015). Sweden was divided in four sub-regions, and for each region, all stations were added to one long time series. From this time series, the POT (Peak Over Threshold) method was applied, and set up such that on average one event were selected per station and year. At least 3 h separation was required





between events for duration less than 3 h, and a separation equal to the duration for longer durations. Then return levels were
derived for several return periods, using the generalized Pareto (GP) distribution fitted using the maximum-likelihood method.

### 2.2.2 Germany

The German DDF statistics are described in (Malitz and Ertel, 2015). The statistics were derived from gauge observations
throughout Germany in the period May to September 1951–2010. A block rainfall method was applied based on the 5-min
base resolution, with adjustments to instantaneous events by multiplication by: 5min - 1.14, 10 min - 1.07, 15 min - 1.04,
20 min - 1.03, and 1.0 for longer durations. A precipitation free time period of at least 4 h between events was required for
durations below 4 h, and a time period equal to the duration for longer durations. POT was applied for sub-daily values, with a
threshold dependent on the lenght of time series such that the threshold is restricted from including more data than the number
of years times 2.718. An exponential distribution was then fitted to the data, and the resulting depths were gridded across
Germany for each given return period. The method is described in the KOSTRA 2010 report (KOSTRA, 2010).

### 2.2.3 Austria

The Austrian data set (Kainz et al., 2007) comes from the Ö-KOSTRA programme, which has many similarities with the
KOSTRA programme from Germany. However, due to a lower number of gauges, the data set is also making use of a convective
precipitation model as support to the gauge analysis. The base resolution is 5-min gauge observations with at least 10–20-year
long records, and the results is a weighted mean of the gauge and model analyses. A POT approach was applied, and more
details can be found in Kainz et al. (2006).

### 2.2.4 The Netherlands

The DDF statistics from the Netherlands are described in (Beersma et al., 2018). The statistics are based on 31 gauge obser-
vations with a 10-min resolution and records of approximately 14 years in the period 2003–2016. All data were pooled and
used as one long time series (436) of annual maxima. The block rainfall approach was used to find annual maxima for different
durations. To accommodate the underestimation introduced when using fixed 10-min intervals rather than instantaneous mea-
surements, a given duration of $t$ min was also considering the $t + 10$ min duration. The generalized logistic (GLO) distribution,
as an alternative to GEV (Generalized Extreme Value) but with a "fatter" tail, was then fitted to the interval of the data with
durations $t$ min and $t + 10$ min. Here, we are using results from Table 2 in STOWA 2018. Since this table lists durations of (1,
2, 4, 8, 12) h and we require also the 3 h and 6 h durations, we derive these by a linear interpolation between 2 h and 4 h, and
4 h and 8 h, respectively.

### 2.2.5 France

The DDF statistics for France were calculated by applying the method SHYPRE (Simulated Hydrographs for flood Probability
Estimation; Arnaud and Lavabre, 2002) to produce rainfall statistics across France (Arnaud et al., 2008). The SHYPRE method





generates data for hourly extremes at a square kilometre scale, from which DDF statistics were derived. This data set is therefore
treated a bit differently regarding the reduction factors, as only the spatial reduction factor is applicable, see Section 3.3.
## 3  Method
### 3.1  Durations
The DDF statistics are derived in a conventional way by employing a running window with a given duration to arrive at the peak
intensity over that window; a so-called "block rain", which does not reflect the actual event durations. We are here confined to
a base resolution of one hour, which means that the one hourly duration is simply taking one hour steps, and no running mean
is possible. This gives an inherent underestimation of the true hourly DDF statistics. For durations above one hour (2, 3, 6, and
12 h are studied), the running window is progressing at one hour steps, giving a steadily more accurate estimate of the peak
intensity.
### 3.2  Extreme value theory approach
Extreme value theory is applied to study precipitation extremes at various durations. Within extreme value theory, there are two
main paths normally taken when it comes to precipitation analyses: annual maxima (AM) or POT (also called partial duration
series (PDS)) (Coles et al., 2001). With the AM approach (often called block maxima) a single event is selected within a block
of data, typically within one year for geophysical time series, and with the POT approach a number of events with values greater
than a given threshold are selected. The latter allows multiple events in a given year to be selected, and additional choices must
be made to assure that the samples are independent and identically distributed (iid). To achieve iid samples, a minimum time
separation is prescribed, such that two events cannot occur too close in time. The time separation varies with the duration such
that for duration below 3 h a minimum separation of 3 h is required, and for duration at or above 3 h, a separation equal to
the duration is required. The selected separation time is set higher than in many studies based on higher temporal resolution
data (e.g. Dunkerley, 2008). Further, it is also set conservatively compared with studies using hourly time steps (Medina-Cobo
et al., 2016) since events are not defined per se, but rather durations, independent on non-precipitation events before and after.
Here, the POT approach is used, mainly because of the 30-year time-slices used for the analysis, for which POT allows a more
robust sample. Pickands-Balkema-de Haans theorem (Pickands, 1975) states that if the samples above the POT threshold are
iid, they will follow a GP distribution:
$$F_{(\xi,\sigma)}(x) = \begin{cases} 1 - (1 + \frac{\xi x}{\sigma})^{-\frac{1}{\xi}} \text{ for } \xi \neq 0 \\ 1 - e^{-\frac{x}{\sigma}} \text{ for } \xi = 0 \end{cases}, \tag{1}$$

where $x > 0$, $\xi$ is the shape and $\sigma$ is the scale parameters. We use Maximum-likelihood for fitting parameters, and return
values are calculated with the inverse cumulative distribution function of a GP distribution with distribution parameters and





probability of exceedance, $p$:
$$p = \left(1 - \frac{1}{T}\right)^{\frac{N}{n}} \qquad (2)$$
where $N$ is the number of records, $n$ is the number of exceedances over the selected threshold, and $T$ is the return period.
There is no well defined method for setting the threshold for POT, but Coles et al. (2001) outlines a method of incrementally
lowering the threshold, i.e. increasing the sample size, and investigating the impact on the parameter fits. Comparing with a
smaller sample, here one event per year on average, the parameters of a larger sample must not deviate beyond the uncertainty
bounds of the smaller sample. We follow Coles et al. (2001) approach as implemented in the R library "extRemes" (Gilleland
and Katz, 2016), and investigate the appropriate threshold for the different durations of one member of the historical period
for each RCM, and in all sub-regions. To determine the threshold at a 95% confidence level, we go through all grid points for
each sub-domain and find the average number of events per year that is rejected by at most 5% of the grid points. The results
are similar over all models, domains and durations, and a threshold of on average three events per year was finally adapted to
all grid points, i.e. a sample size of 90 events for each extreme value fit. Comparisons using the Gumbel distribution calculated
from annual maxima gave very similar results for the ten year return values, although with more spatial variability (noise),
which is most likely due mainly to the smaller sample size.
## 3.3   Comparison across spatio-temporal scales
To evaluate the model simulations, DDF statistics were collected from different national authorities across Europe. Most of
these data sets are based on gauge data at minute scale temporal resolution, which is inherently different from the about 12 km
and one hourly data of the models (e.g. Eggert et al., 2015; Haerter et al., 2015). A direct comparison would reveal a biased
comparison where gauge based data have significantly higher return values due to their better sampling of the peak of a given
duration window, as well as the peak within a precipitation area.
To alleviate this bias, we first derive area and time reduction factors that can be applied to each local data set. We make use
of the Swedish radar and gauge based data set HIPRAD (Berg et al., 2016) as well as 15 min resolution gauge records for the
same domain, to derive time and areal reduction factors based on annual maxima for the years 2011–2014, see Table 2. Some
grid points, primarily in northern mountainous regions of Sweden, were masked out from the analysis due to unrealistic data. In
Olsson et al. (2018b), the intensity reduction for hourly aggregations between near instantaneous and 15 min gauge resolution
data was studied with Swedish records and found to be about 4% at the one hourly durations and negligible at 6 h duration.
HIPRAD is originally available at a 2 km grid and 15 min resolution, and was used to compare the reduction factors when
both time and space coarsening is considered. When coarsening the time and space resolutions from 2 km and 15 min data to
$0.11°$ and 60 min data, the reduction is about 16% at hourly duration and falling to only about 1% at 12 h duration. The final
conversion factor to go from a near instantaneous point source rain gauge measurement to the 1 h and $0.11°$ resolution model
data becomes the product of the time reduction factor of the gauge data and the space and time reduction factor of HIPRAD,
as shown in the last line of Table 2. These factors compare well to previously applied area reduction factors (Sunyer et al.,
2016), e.g. (Wilson, 1990) presented a factor 1.279 for hourly precipitation, although at 24 h duration the factor only decreased



**Table 2.** Relative differences in annual maxima averaged over four years at different temporal and/or spatial resolutions.

| Data1 | Data2 | 1h | 2h | 3h | 6h | 12h |
|---|---|---|---|---|---|---|
| Gauge(point; instant) | Gauge(point; 15 min) | 1.04 | 1.03 | 1.02 | 1.00 | 1.00 |
| HIPRAD(2 km;15 min) | HIPRAD(0.11°;60 min) | 1.16 | 1.06 | 1.04 | 1.02 | 1.01 |
| HIPRAD(2 km; 60 min) | HIPRAD(0.11°;60 min) | 1.03 | 1.02 | 1.02 | 1.01 | 1.00 |
| Final Reduction factors | | 1.21 | 1.09 | 1.06 | 1.02 | 1.01 |

to 1.066 indicating a slightly too small factor in our current study. Such differences can be explained by differences in local precipitation climate, and is regarded as an inherent uncertainty in this analysis. The factors are applied to the gauge based local data sets, and for the French SHYPRE data set, only the space reduction factor for 60 min duration is applied.

## 4 Results

### 4.1 Evaluation

Due to the different methodologies applied in the different national data sets, the evaluation is mainly considering the 10-year return level, as this is well within the sample coverage of the data series and is therefore not so sensitive to the choice of method for extreme value calculations, e.g. considering the use of AM or POT, or the extreme value distribution applied. A general overview of the parameters fits of the extreme value distribution shows minor influence of the driving GCM, but there are differences between the RCMs. At 12 h duration all RCMs have similar parameter values across Europe (see Fig. S1 and Fig. S2), but at 1 h duration there are more regional differences, and especially RACMO22E differs with a lower scale parameter (see Fig. S3 and Fig. S4). The differences in the GP parameters indicate differences in the mean and variance of the events in the different RCMs, which might be due to, e.g., grid point storms at short durations as pointed out by Chan et al. (2014).

When evaluating the DDF statistics, the reduction factors of Table 2 were applied to all national data sets, except for France where the scale gap in time is inherently bridged and only the space scale is adjusted, see Section 3. Figure 1 presents the evaluation results for each of the domains with local data. Since only GCM driven simulations have been analysed, the evaluation is not purely of the RCMs as would be approximated in reanalysis driven simulations, but of a mixture between the driving GCM and the RCM response to that forcing. Still, RCM dependent impacts can be seen in the results. For all domains and most models there is a clear pattern of large dry bias for 1 h duration, with a clear decrease in bias with longer durations. The main exception from this is the REMO2009 model which agrees better with observations across all durations. Also HIRHAM5 is performing better than the RCA4 and RACMO models, however with a wetter bias for longer durations. The RACMO22E model produces strong underestimations of extreme intensities, mostly between about -25 and -50%.

Observation based data sets over Germany and France are available as maps, making a visual evaluation possible. Figure 2 and Fig. 3 show the 10-year return level for one and 12 h durations over Germany, respectively. For both presented durations,



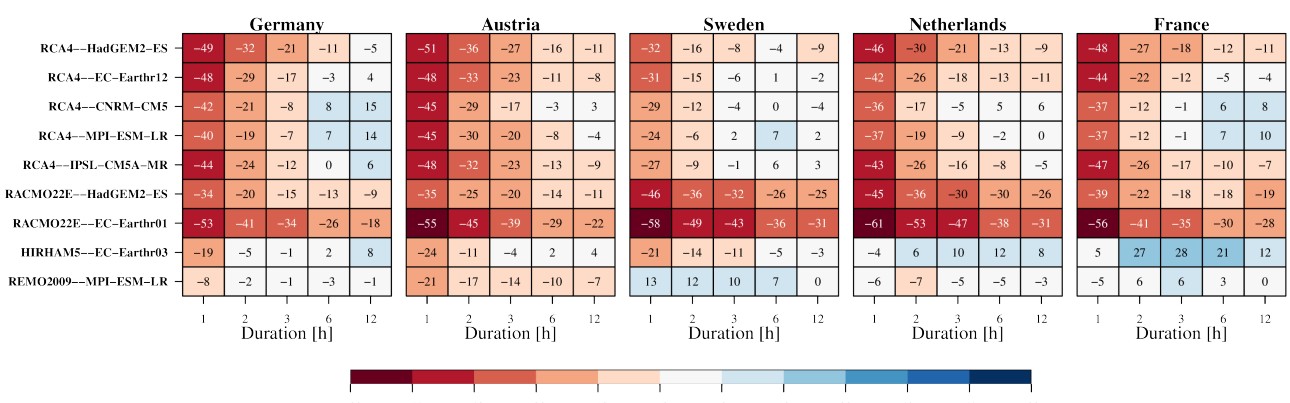

**Figure 1.** Evaluation of model ensemble for selected regions and for the 10-year return period. Gauge based observations have been adjusted for spatial resolution and time sampling to approximate the statistics of the model resolution and sampling as explained in the main text. Both colours and numbers indicate the bias.

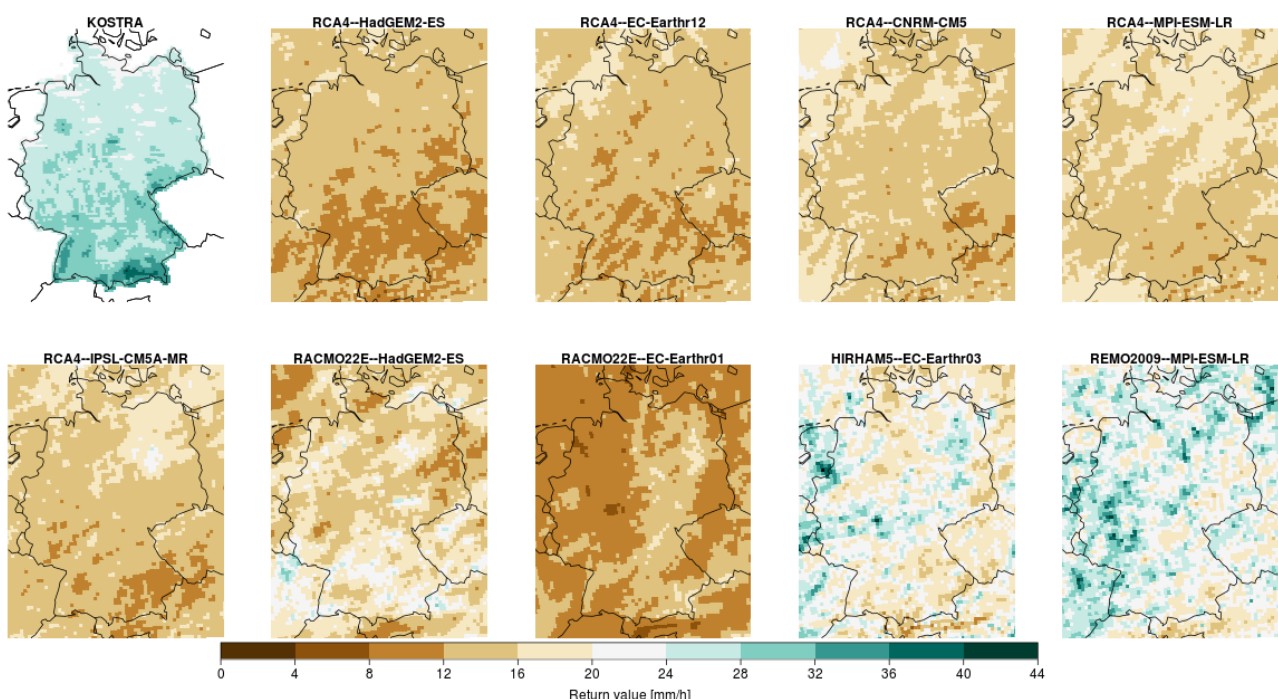

**Figure 2.** 10-year return level for 1 h duration of KOSTRA and all models in the RCM ensemble for Germany.


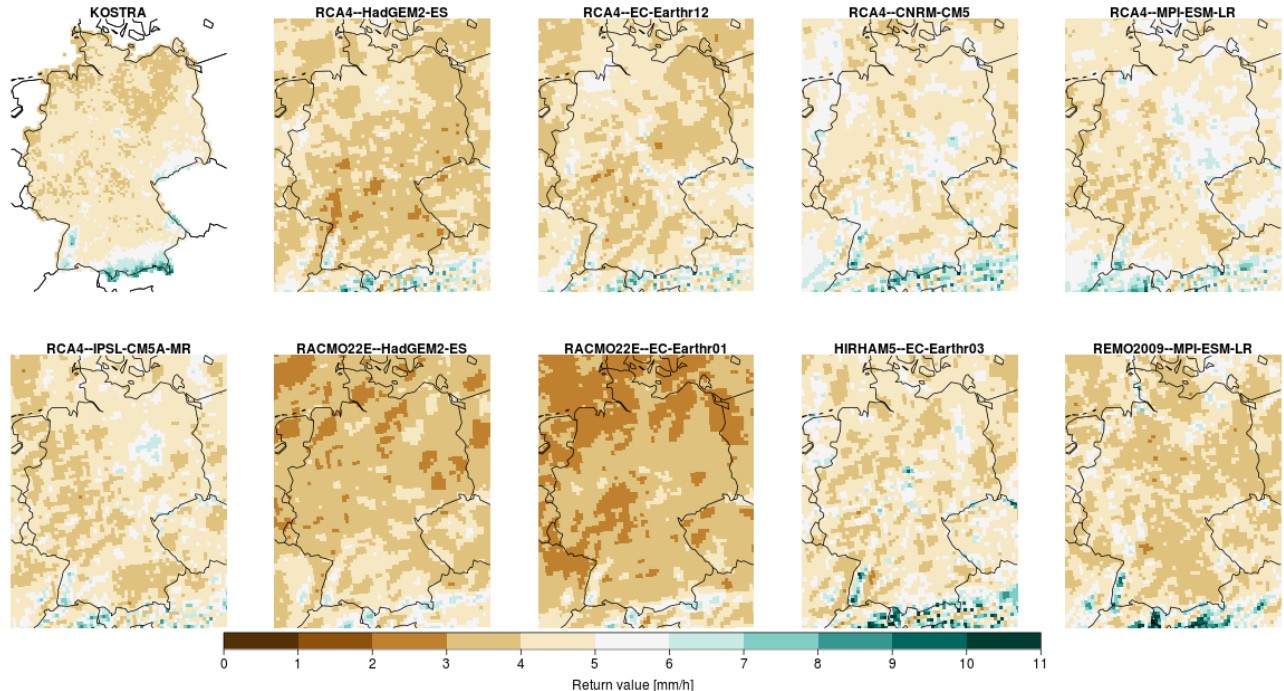

**Figure 3.** 10-year return level for 12 h duration of KOSTRA and all models in the RCM ensemble for Germany.

the observations show two main high intensity regions in Germany: one in the pre-Alpine area close to the south-eastern
border to Austria, and one in the Black forest region oriented in north-south direction in the south-west. Intensities tend also to
decrease towards the north. For the hourly duration, all but HIRHAM5 and REMO2009 severely underestimate the intensity,
as seen also in Fig. 1. Here, we see that they also fail in reproducing the spatial pattern, especially for RCA4 which fails to
reproduce both the orographic regions in the south, and also a reversed north-south gradient. Further, the maps for HIRHAM5
and REMO2009 clearly show that although these two simulations perform better in the median intensities in Germany they
also fail in reproducing the spatial pattern. The spatial analysis shows that the better performance derived from Fig. 1 is due to
generally higher precipitation intensities of the REMO2009 and HIRHAM5 RCMs, but not in the right locations. Only when
increasing the duration to 12 h do the models start to reproduce the observed spatial patterns, see Fig. 3.
Figure 4 and Fig. 5 show similar maps for France and the observation based data set SHYPRE. SHYPRE shows the highest
intensities along the Mediterranean coastline and over the island of Corsica, and intensities decrease gradually towards the
north-west. As for Germany, all models but HIRHAM5 and REMO2009 generally underestimate one hourly intensities, and
the peak intensity region is poorly reproduced in RCA4, and only somewhat better in the RACMO22E simulations. Within
the ensemble of each individual RCM, there are variations that are likely due to the driving GCM, however, these variations
are small compared to the intra-RCM spread. HIRHAM5 and REMO2009 have clear intensity maxima in the south of France
that reproduces well that of SHYPRE. Twelve hourly durations are better simulated by all models, with at least the general




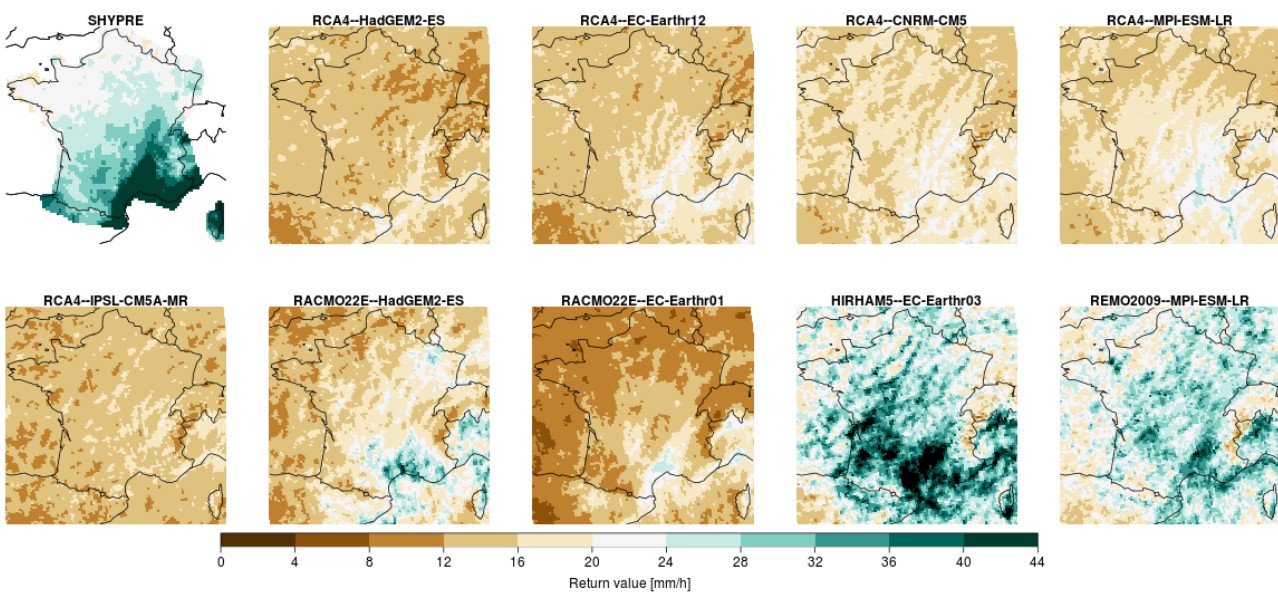

**Figure 4.** 10-year return level for 1 h duration of SHYPRE and all models in the RCM ensemble for France.

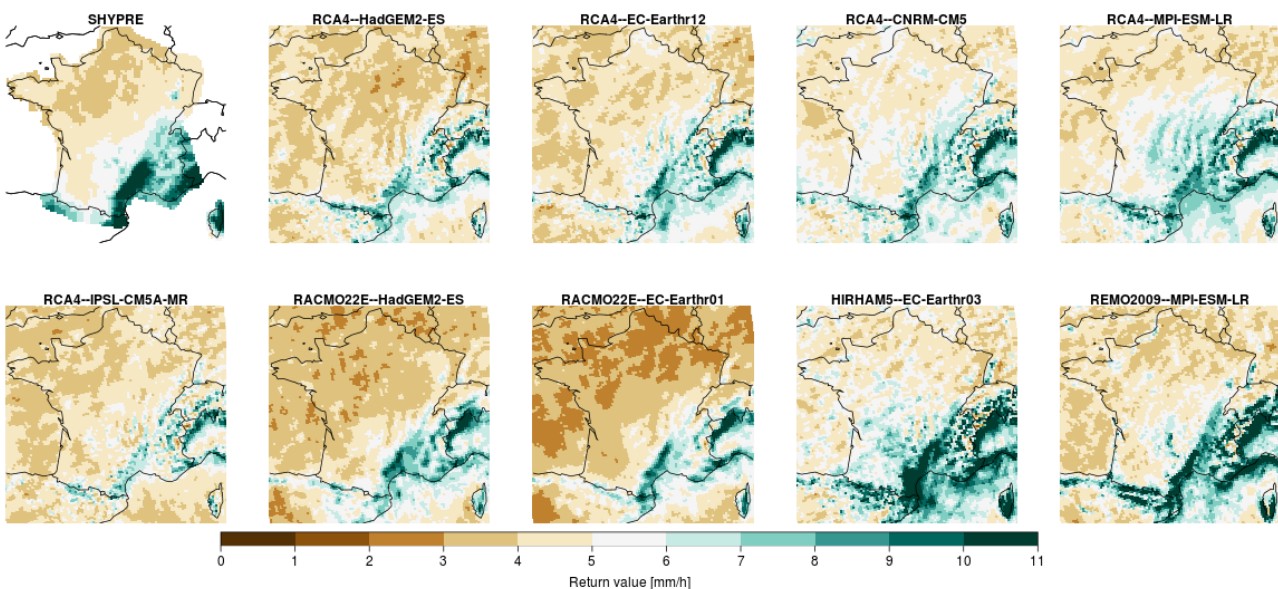

**Figure 5.** 10-year return level for 12 h duration of SHYPRE and all models in the RCM ensemble for France.

1   pattern similar to SHYPRE. However, RCA4 and RACMO22E are still underestimating intensities, whereas HIRHAM5 and

2   REMO2009 show better agreement regarding intensities.



Natural Hazards
and Earth System
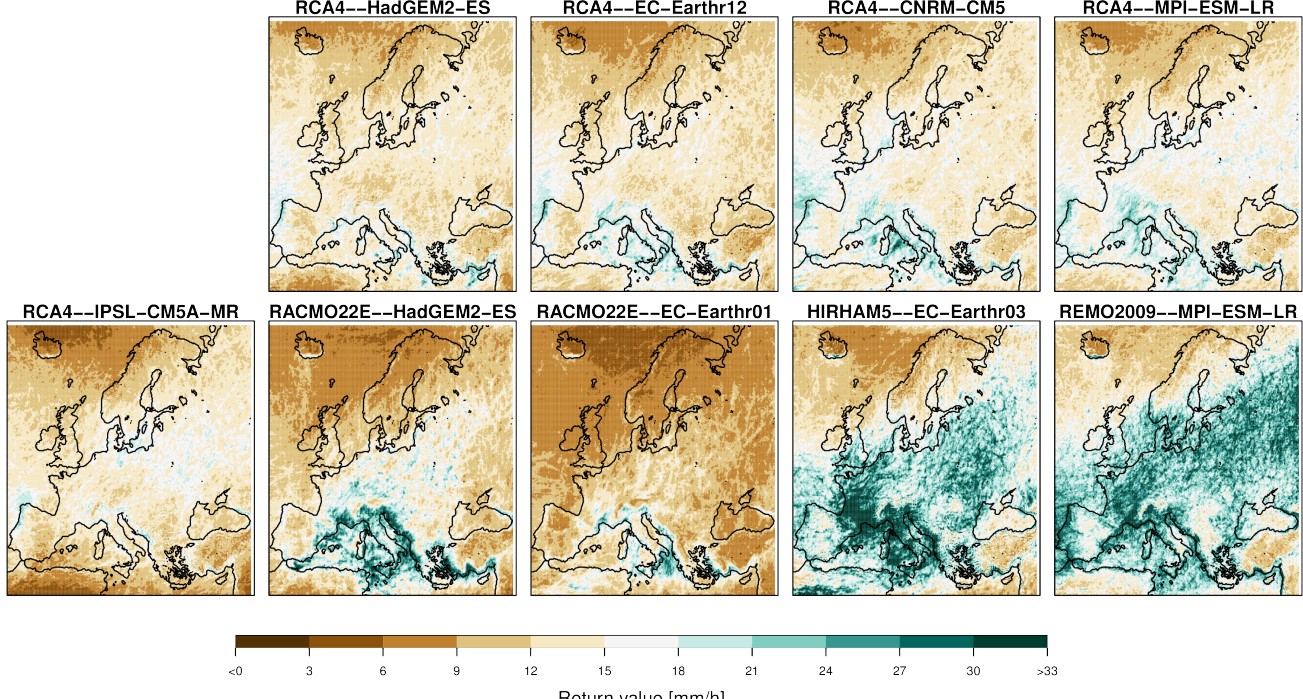

**Figure 6.** 10-year return level for 1 h duration of all models in the RCM ensemble.

To complement the evaluation with a pan-European view of modeled extreme intensities, Fig. 6 and Fig. 7 show the 10-year
depths for one and 12 h durations, respectively. At 1 h duration, all models share a similar structure of higher intensities over
the ocean west of France and the Iberian Peninsula, and along the northern Mediterranean coastline; although the magnitude
differs between the models. The different RCA4 simulations show that the driving GCM has some impact on the pattern
across Europe. For example, HadGEM2-ES produces less intense rainfall in southern France, where the MPI-ESM-LR driven
simulation has generally more intense rainfall. However, the driving GCM seems to have less influence than the RCM. At
12 h duration, the general patterns across Europe converge across all GCM-RCM combinations, although with differences in
overall intensities, see Fig. 7. However, it is unclear from this study whether the pattern is correct or not, since observations
are lacking. Earlier studies have indicated that the core of the events are underestimated by the parametrised 0.11° simulations
(Kendon et al., 2014), but the large bias in the 1 h durations might also indicate that small concentrated events are missing from
the parametrised simulations.
The general conclusion is that hourly durations are underestimated in the models, which is a likely consequence of model
resolution and deficiencies in convective parametrisations. Longer duration events which also tend to have a larger spatial
extent are better captured by the grid resolved component of the model simulations, where also orographic effects become
more clear in the spatial patterns, in agreement with observations.





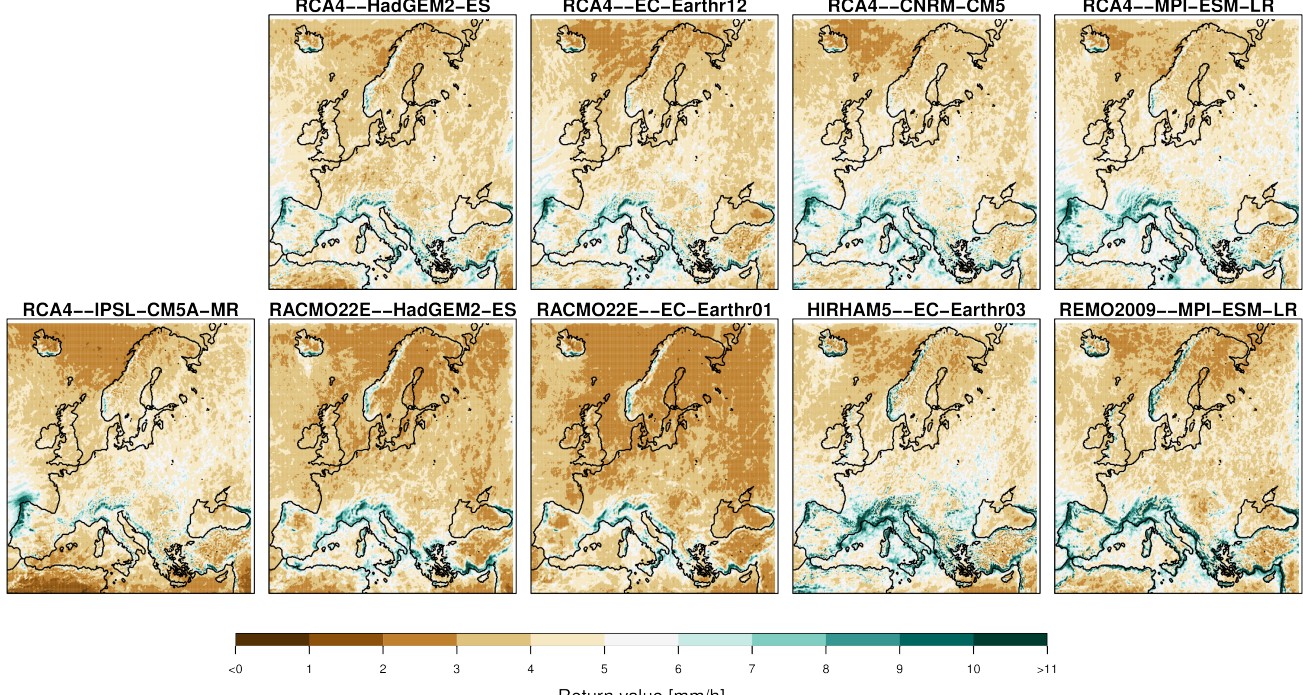

**Figure 7.** 10-year return level for 12 h duration of all models in the RCM ensemble.

## 4.2 Future projections

The performance of the RCMs in reproducing observed patterns for 12 h durations is promising enough to promote further analysis of future projections. We include also shorter durations in the analysis, despite their poor evaluation performance. Here, we investigate the response of extreme precipitation as a function of the local summer half-year (April–September) temperature change in three future time slices: 2011–2040, 2041–2070, and 2071–2100. The analysis is performed at land-points for the so-called PRUDENCE regions (BI=British Isles, IP=Iberian Peninsula, FR=France, ME=Mid-Europe, SC=Scandinavia, AL=Alps, MD=Mediterranean, EA=Eastern Europe; Christensen and Christensen, 2007), and the depths are related to the change in mean temperature for each sub-region, between the future time slices and the historical reference period 1971–2000.

Figure 8 shows scatter plots of the changes in 10-year depths for precipitation of 12 h duration, with the change in local summertime temperature for each ensemble member. The relative change in precipitation was calculated by first performing a domain average, and then calculating the change between time periods. First, it is clear that the scatter plots have strong linear trends even when considering different sub-regions, different time slices and different emission scenarios. This indicates a strong connection between the change in precipitation extremes and the seasonal temperature. Second, the individual RCMs show large differences in their response depending on the driving GCM, but also different RCMs respond differently to the same GCM. Results for 1 h duration show larger spread, but still good linear fits, and stronger scaling (see Fig. S5).



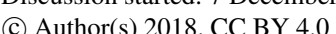


**Figure 8.** Scatterplot of the relative change in 10-year 12 h depths against summertime mean temperature change between future and historical time periods, for different sub-regions, emission scenarios, and time periods according to the legend. Each panel show the result for different RCM-GCM combinations. Linear fits to all data are presented in each panel, along with slope and intercept coefficients as well as the R-square value of the fit. CC-rate changes of 7%/K are shown as gray lines in the plots.



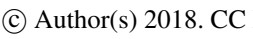

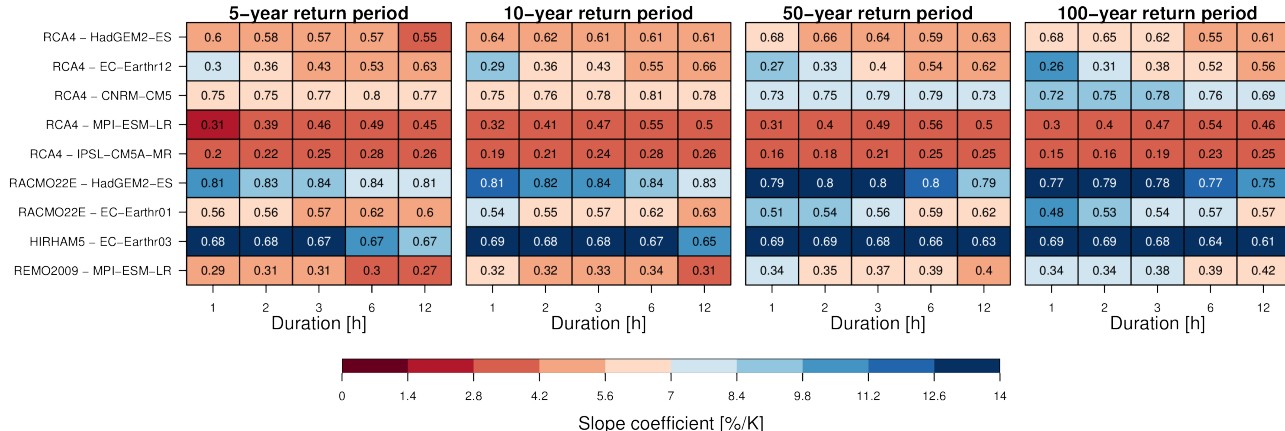

**Figure 9.** Summary of the relative change in precipitation extremes (5, 10, 50, 100 year depths), at various return periods, against summertime temperature change between future and historical time periods for all PRUDENCE regions and RCPs and time-slices together. The displayed changes are calculated as the slope coefficient of linear fits as in Fig. 8. The colour scale is set relative the Clausius-Clapeyron prediction of about 7%/K, with red (blue) colour showing scaling below (above) that rate. The numbers in each box presents the R-square value for the individual fits as a measure of the goodness-of-fit.

To further investigate the connection between extreme precipitation and seasonal temperature, we perform linear fits for
each RCM-GCM combination, see Fig. 8. The results are summarized for all durations and return periods in Fig. 9, with colour
coding such that increases beyond the CC-rate are in shades of blue, and below the CC-rate are in shades of red. All model
combinations show a positive relationship, i.e. increasing slopes, but the slopes vary between about 1 to over 10%/K. Most
model combinations show stronger scaling for shorter durations (towards the left in each panel) and an increase in the scaling
with increasing return period (panels toward the right in Fig. 9). The exceptions are the models RCA4-MPI-ESM-LR and
RCA4-IPSL-CM5A-MR which remain fairly constantly around 3%/K scaling for all durations and return periods. Comparing
the influence of the RCM, it is interesting to see that RCA4 driven with EC-Earth scales stronger than with HadGEM2-ES,
whereas the opposite is the case for RACMO22E, although the realisation of EC-Earth is different, which might have an
influence that we cannot quantify in this study. REMO2009-MPI-ESM-LR has slightly stronger scaling than RCA4-MPI-
ESM-LR, and HIRHAM5-EC-Earth scales much stronger than the RACMO22E and RCA4 simulations with the same GCM.
## 5  Discussion
Sub-daily Precipitation measurements are performed throughout Europe; partly in country wide organized way by the mete-
orological offices, but also frequently by local counties. Access to these data are mostly restricted, or simply impractical at
larger scales, although initiatives such as the INTENSE project has come a long way in collecting such data (Blenkinsop et al.,
2018). National DDF statistics are often available in some form, and a detailed inventory of these data sets would be a valuable





first step in collecting a Europe wide data set for evaluating model simulations. A first step was taken in this study, but a closer
involvement of the data providers would be necessary to assess details of the sometimes cryptically explained data processing
methods. A further complication is that most national data sets are decribed only in the native language.
The national DDF data sets were here employed as qualitative indicators for the performance of RCM simulations. Some
challenges with comparing DDF statistics are due to how they were derived: using different methodologies, gauge resolution
and record lengths, mixes of observations and model data, etc. The evaluation was therefore restricted to the 10-year return
period, which is shorter than the gauge record lengths in all data sets and therefore less dependent on the employed extreme
value estimation method. More in-depth analysis would require a larger undertaking in comparing the implications of every
choice made in the different data sets and how they affect the final result. A spatial evaluation of the RCMs was performed for
the German and French data sets, and also here, only the main patterns connected to known physical processes are discussed
due to large uncertainties.
The four RCMs in the investigated model ensemble show significant differences in the simulations of extreme sub-daily
precipitation. This is in spite of the similarities of several of the models. For example, the convective parametrisation is similar
for HIRHAM5, REMO2009 and RACMO22E, which are all based on Tiedtke (1989), however, with differences in their
settings and in additions to the parametrisations. Further, HIRHAM5, RACMO22E and RCA4 share similar dynamical cores
(originating from the HIRLAM NWP model). Still their responses are quite different when it comes to extreme precipitation
and their response to future emission scenarios. This emphasizes the importance of the complete set of parametrisations and
parameter sets in the models.
Differences in settings within the convection schemes, such as the mass flux closure used, can have significant impact. Also
other parametrisations such as turbulence scheme, surface roughness settings or smoothing of the orography can significantly
affect the mixing in the lower boundary and thereby affect the sensitivity of convective triggering. The effects of the parametri-
sations can feedback with the dynamics of the model, and produce highly non-linear responses. Thus, reducing the fully three
dimensional processes into simplified one or two dimensional parametrisations is indeed challenging. The separation of the pre-
cipitation process into resolved and un-resolved (parametrised) components is especially problematic for cloud bursts, where
large scale moisture convergence is present and can lead to positive feedback through latent heat release (Lenderink et al.,
2017; Nie et al., 2018).
An important result is the apparently good performance of the RCMs HIRHAM5 and REMO2009 on domain average
statistics, whilst a closer look at spatial patterns reveals an actually poor performance. More data of DDF statistics across
geographical domains is essential for model evaluation, and we call out for more national institutes to open up their records
and share their statistics. For example, domain average DDF statistics over the Alps region presented in Ban et al. (2018) show
fairly equal performance at 12 km and 2 km resolution. However, domain averaging might hide important differences between
model simulations, which could inform about the different models' actual performance.
Scaling of precipitation extremes with future projections are here studied by comparing relative changes in precipitation
intensities as a function of surface temperature increase. Recently, Ban et al. (2018) performed a similar study relating seasonal
mean temperature and precipitation changes, with the result that both the 0.11° and 2 km simulations agree on a close to 7%/K



scaling. When set into context of the current study, we see that this result might be influenced by both the choice of RCM and GCM, stressing the importance of ensembles also for kilometre scale studies.

# 6 Conclusions

Extreme precipitation at sub-daily time scales in the summer half-year are investigated with a EURO-CORDEX ensemble at 0.11° resolution. The extremes are estimated using a POT approach with a GP distribution, and the results are evaluated against national information for several countries across Europe. From the evaluations, we conclude that:

- All models perform poorly at hourly duration, with increasing performance for longer durations.

- Spatial patterns are reasonably well represented only at 12 h duration, indicating a disconnect between orography and extreme events at shorter duations.

- Both the GCM and RCM affect both magnitudes and spatial patterns across Europe, but the RCM is most prominent in shaping the spatial structure at short durations.

Future projections are investigated through a connection with summer half-year mean temperature and precipitation change for the time-slice periods 2011–2040, 2041–2070 and 2071–2100. The results are presented as %/K changes, and we conclude that:

- The %/K-scaling works well across sub-regions, time-slices and RCP scenarios, such that all aligns practically linearly.

- The scaling display a large spread between models, with about equal impact of the GCM and the RCM.

- Scaling of extreme precipitation with temperature is positive across the model ensemble, resulting in an ensemble mean slightly below the CC-rate, but ranges from about half to about two times the CC-rate for different ensemble members.

The concept of relating extreme precipitation changes to temperature seems to be a valid and useful approach to predict changes in extreme precipitation. However, this conclusion might be a bit rash since the performance of the models is poor for short durations and do not inspire trust in their application for future projections. The next generation of convection permitting models might perform better, but their improved performance in reproducing the spatial pattern of extreme precipitation across domains should be investigated. For this, we urge national authorities to openly and transparently share assessments of DDF statistics from their high resolution observations.

# 7 Data availability

The hourly EURO-CORDEX data are not part of the standard suite of CORDEX, and are therefore not available from the ESGF-nodes and are not produced nor shared by all model groups. The existing data can be accessed upon request from each model institute, on their good will and capability.



*Author contributions.* PB and JO designed the DDF calculation strategy. WY calculated the DDFs. PB conceptualized the evaluation and
future projection analysis and performed the statistical evaluation. KK performed the analysis on future projections. GL, OBC, and CT
contributed with model specific insight. PB prepared the manuscript with contributions from all co-authors.
*Competing interests.* The authors declare that they have no conflict of interest.
*Acknowledgements.* This work was in part funded by the projects SPEX and AQUACLEW, which is part of ERA4CS, an ERA-NET initiated
by JPI Climate, and funded by FORMAS (SE), DLR (DE), BMWFW (AT), IFD (DK), MINECO (ES), ANR (FR) with co-funding by
the European Commission [Grant 690462]. We acknowledge the provision of DDF statistics from the various national athorities, and our
colleagues around Europe that help in collecting the data sets for us.



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
