# Peer review of "Summertime precipitation extremes in a EURO-CORDEX 0.11° ensemble at hourly resolution"

_Natural Hazards and Earth System Sciences, 2018_

## Referee Comment (RC1) · Anonymous Referee #1 · 2 Jan 2019

The paper "Precipitation extremes in a EURO-CORDEX 0.11° ensemble at hourly resolution" by Berg et al. presents the difficulties and uncertainties regarding the evaluation an future projections of sub-daily precipitation extremes and how far they can be tackled at the mement. The topic is relevant and suitable for this journal. The manuscript is well written and the overall quality of the presentation is good. It offers new insights into the topic. They demonstrate how unsatisfactory the data availability for sub-daily precipitation observations as well as high-resolution simulation data is at the moment. The make the effort to compile the observational base at least for several European regions and compare the respective methods to derive the extreme precipitation depth-duration-frequency functions (DDF). The results of the evaluation and the large ensemble variability prove the necessity to examine this topic further including

more simulations. At this stage neither the quality of the RCMs to reproduce short term precipitation extremes or the questions regarding the CC-scaling in the future can be fully determined, as the manuscript shows in a convincing way. I fully agree with the authors, that sub-daily data should be added to the ESGF, if possible for the existing CORDEX simulations but certainly for future efforts. This is not only necessary for precipitation but also for other variables like e.g. wind. I recommend this paper for publication. Additionally, efforts to provide suitable comparable Pan-European observation data for the analysis of extremes and model-evaluation are highly appreciated. There is just a minor questions/suggestions: - The authors derive the DDF for five regions. The spatial distributions are shown just for Germany and France. Is the representation of the spatial pattern comparable in the other regions? Due to the different methodologies used to derive the DDF it is difficult to distinguish which of effects presented stem from the differences of the methodologies and which from the problems of the models to represent short term extremes.

---

## Referee Comment (RC2) · Anonymous Referee #2 · 4 Jan 2019

General comments:

The representation of sub-daily precipitation extremes and their future changes are investigated using a subset of EURO-CORDEX 0.11° climate models. The article gathers an impressive number of datasets and hourly-output from models to assess the limits to the use of convection-parameterised models at sub-daily time-scales in summer, which had never been done before. The authors first provide an evaluation of depth-duration-frequency curves (return-levels) against pre-calculated country-wide DDF curves. They conclude that convection-parameterised models at 0.11° are not able to represent hourly intense rainfall events: they mostly underestimate 10-year return level precipitation. Their ability is mostly RCM dependent. However, the models show skills in representing 12-hourly return values. The authors show that the 12h

return value is increasing with temperature in future climate scenarios, but that the slope depends both on the RCM and the GCM. Although not reliable, hourly intensities increase generally at a larger rate than 12-hourly intensities. This study introduces an interesting methodology and comparison with observations which could be further used in the assessment of future convection-permitting ensembles of models.

I find the article scientifically robust, written in a clear manner and worth of publication in NHESS. I mainly have minor comments, which I believe could improve the manuscript.

Specific comments:

1) P6L19-21: Is the 3h separation for values below 3h enough to assume "iid"? You write that this is higher than many studies, but it is lower than Ban et al (2018) (2days) or Chan et al. 2014 (1day). Does using 1 day for all durations significantly impact the results? L21-22 is not comprehensible. Please clarify.

2) P4 L17: "The analysis is restricted to summer-half years (April–September) to focus on the main convective season in Europe (Berg et al., 2009)." Note that you are missing most of the season of deep convective events in the Mediterranean (Sept.-Dec.): it may be worth producing the French map or Europe-wide map for this season, or extending the season to October. e.g.: Enno, Sugier and Alber (2018) Lightning flash density in Europe on the basis of 10 years of ATDnet data; 25th international lightning detection conference & 7th international lightning meteorological conference You could also note that seasonality changes, such as reported by Marelle et al. (2018) are not taken into account in your study.

3) This is a semantic question, but I find the term "cloud burst" in the introduction rather ill-defined, it seems to be defined by its impacts, and to correspond to convective rainfall above 100mm/h? 50mm/h? 12h duration rainfall is probably more like frontal rainfall in most european regions, is this a "cloud burst"? I would use the term "heavy precipitation event" or extreme precipitation event, which is probably less dependent on the type of precipitation event/the impacts it has.

4) P2L34-35: Ban et al. (2018) do not find a stronger scaling for intense events in convection-permitting models compared to convection-parameterised models, to the contrary it is weaker in summer, which is your season of interest.

5) Figures 2-7 and S1-4: you show continuous fields with a diverging color-bar, this can me a bit misleading, please use a sequential (multi-hue) colorscale. https://journals.ametsoc.org/doi/pdf/10.1175/BAMS-D-13-00155.1 You could start the colorscale above 0 to use all the colour intervals in the figure.

6) It'd be interesting to see the spatial variability of the precipitation enhancement thanks to the additon of a map of future changes (e.g. 10-year return value of 12h-duration) for RCP8.5. In Fig. 8, you are pooling the results in a single figure, on which it is difficult to see individual regions (you could reduce the y limits to 60% (or 90Âǎ% if you want to keep consistency between hourly and 12-hourly graphs).

7) P2L20-21: you could add that convection-permitting models better represent Mediterranean heavy precipitation events (which stand out in your Fig. 4-5) and in some regions still overestimate moderate to intense hourly precipitation (Berthou et al. 2018).

Technical corrections:

P2L16: add "in Sweden". P8L9: parameters fits -> parameter fits P10L15: intra-RCM spread -> inter-RCM spread P12L9: the core of the events -> the peak of the events P6L24: de Haans -> de Haan P6L24: Picklands (1975) not referenced

Berthou, S., Kendon, E. J., Chan, S. C., Ban, N., Leutwyler, D., Schär, C., & Fosser, G. (2018). Pan-European climate at convection-permitting scale: a model intercomparison study. Clim. Dyn. http://doi.org/10.1007/s00382-018-4114-6

Chan, S. C., Kendon, E. J., Fowler, H. J., Blenkinsop, S., & Roberts, N. M. (2014). Projected increases in summer and winter UK sub-daily precipitation extremes from high-resolution regional climate models. Environmental Research Letters, 9(8), 84019.

Marelle, L., Myhre, G., Hodnebrog, Ø., Sillmann, J., & Samset Bjørn, H. (n.d.). The changing seasonality of extreme daily precipitation. Geophysical Research Letters, 0(ja). http://doi.org/10.1029/2018GL079567
* * *

---

## Referee Comment (RC3) · Anonymous Referee #3 · 24 Jan 2019

The manuscript of Berg et al. provides a comparison between regional climate model outputs of precipitation and high-resolution observational datasets in Sweden, Germany, Austria, Netherland and France. Overall, the manuscript is well written, the objectives are clear and the results support the goals of the study. Yet, I am puzzled with this submission since to my opinion it does not bring new results. Indeed, the conclusion can be found in the introduction, page 2, line 13-19:

"However, RCMs and GCMs have shown severe problems with their sub-grid scale parametrisations of convective processes, which affect their ability to reproduce, e.g., the diurnal cycle of rainfall intensity (Trenberth et al., 2003; Fosser et al., 2015; Prein et al., 2015), the peak storm intensities (Kendon et al., 2014), and extreme hourly intensities (Hanel and Buishand, 2010). It is therefore questionable to which extent such

[Figure]

RCMs are capable of describing cloudbursts in present as well as in future climate"

Indeed, it is well known that the current generation of CORDEX RCMs includes a convective scheme that is not able to reproduce adequately the small-scale high-intensity rainfall events. Beranová et al. (2018) evaluated the hourly outputs of RCMs and projections for short duration's rainfall have been provided by Kyselá et al. (2012), among others. This is the reason why regional climate models that explicitly reproduce convection are being developed, there is a huge amount of literature presenting this new generation of climate models, see for instance Coppola et al. 2018 or Berthoux et al. 2018 (I believe both should be cited in the text). However, I agree as stated by the authors page 3, line 1 that the convection-permitting simulations are still not widely available, unlike EuroCordex runs. Yet, when reading the manuscript it seems that these convection-permitting simulations are still not available for research purpose, when several studies have already been produced with these types of model (see Berthoux et al. 2018, Reszler et al. 2018). It can be somewhat misleading to the reader not familiar with climate models.

Specific comments:

Since the study focuses on the summer season, the title should say it. In various regions such as south France, the maximum intensity events are occurring in the autumn, not during summer.

Page 4, line 6: Rajczak and Shär 2017 analyzed daily model outputs

Page 3 section 2.1: it should be clearly stated here that the 9 simulations all include a parametrized convection scheme.

Page 13, lines 9: it is not clear which threshold is used in the GP model for future time periods. As explained page 7, lines 7-14, a precipitation threshold is defined for each grid point to have 3 events on average per year. Which value is used for the future time period? the threshold value yielding 3 events per year in present climate ? The

authors should provide, at least in the text, the ranges of threshold values obtained for the different grid points/regions.

Page 15, line 13-15: it is very good that the authors talk about data availability in the discussion. It should be stressed also that the different data sets they used are probably not homogeneous at all: some rely on observed precipitation, some rely on a mixture of observed precipitation and simulations from a climate model (Germany) and some rely on a weather generator (France). Further work should try to homogenize these data sets prior to the evaluation of climate models, or the discrepancies between data set could induce an artificial bias in the evaluations. Due to different sources of data, is it very likely that the spatial patterns of the different datasets cannot be compared in a robust way.

References:

Berthou, S., Kendon, E. J., Chan, S. C., Ban, N., Leutwyler, D., Schär, C., and Fosser, G.: Pan-European climate at convection-permitting scale: a model intercomparison study, Clim. Dynam., https://doi.org/10.1007/s00382-018-4114-6, in press, 2018

Beranová R., Kyselá J., Hanel M., 2018: Characteristics of sub-daily precipitation extremes in observed data and regional climate model simulations. Theoretical and Applied Climatology, 132, 515-527

Coppola, E., Sobolowski, S., Pichelli, E., Raffaele F., Ahrens, B., Anders, I., Ban, N., Bastin, S., Belda, M., Belusic, et al., 2018. A first-of-its-kind multi-model convection permitting ensemble for investigating convective phenomena over Europe and the Mediterranean. Climate Dynamics https://link.springer.com/article/10.1007/s00382-018-4521-8

Kyselá J., Beguería S., Beranová R., Gaál L., López-Moreno J.I., 2012: Different patterns of climate change scenarios for short-term and multi-day precipitation extremes in the Mediterranean. Global and Planetary Change, 98-99, 63-72

Reszler, C., Switanek, M. B., and Truhetz, H.: Convection-permitting regional climate simulations for representing floods in small- and medium-sized catchments in the Eastern Alps, Nat. Hazards Earth Syst. Sci., 18, 2653-2674, https://doi.org/10.5194/nhess-18-2653-2018, 2018.
* * *

---

## Author Comment (AC1) · 13 Mar 2019

**We appreciate very much the comments and suggestions, as well as the time and energy spent in reviewing our manuscript. Below are answers to all items raised.**

The paper "Precipitation extremes in a EURO-CORDEX 0.11âŮę ensemble at hourly resolution" by Berg et al. presents the difficulties and uncertainties regarding the evaluation an future projections of sub-daily precipitation extremes and how far they can be tackled at the mement. The topic is relevant and suitable for this journal. The manuscript is well written and the overall quality of the presentation is good. It offers new insights into the topic. They demonstrate how unsatisfactory the data availability

[Figure]

for sub-daily precipitation observations as well as high-resolution simulation data is at the moment. The make the effort to compile the observational base at least for several European regions and compare the respective methods to derive the extreme precipitation depth-duration-frequency functions (DDF). The results of the evaluation and the large ensemble variability prove the necessity to examine this topic further including more simulations. At this stage neither the quality of the RCMs to reproduce short term precipitation extremes or the questions regarding the CC-scaling in the future can be fully determined, as the manuscript shows in a convincing way. I fully agree with the authors, that sub-daily data should be added to the ESGF, if possible for the existing CORDEX simulations but certainly for future efforts. This is not only necessary for precipitation but also for other variables like e.g. wind. I recommend this paper for publication. Additionally, efforts to provide suitable comparable Pan-European observation data for the analysis of extremes and model-evaluation are highly appreciated. There is just a minor questions/suggestions: - The authors derive the DDF for five regions. The spatial distributions are shown just for Germany and France. Is the representation of the spatial pattern comparable in the other regions?

**We show only the spatial distributions for Germany and France, as we only have gridded data for those two observational data sets. However, we also show the RCM spatial patterns for the full model domains, which shows at least the differences between the models in different regions. E.g., REMO shows similar low correlations with orography in Sweden (compare the very southern parts and the Scandic mountains bordering Norway in Fig. 6). This is stated on Page 8 line 24, but we will stress this point for each data set in Section 2.2 in the revised manuscript.**

Due to the different methodologies used to derive the DDF it is difficult to distinguish which of effects presented stem from the differences of the methodologies and which from the problems of the models to represent short term extremes.

**Indeed, this is an issue. This is why we focus on the 10-year return period that is**

within the data range of the different data sets. In our experience, this reduces the impact of the methodological choices for the extreme value analysis, as we argue for on Page 8 lines 6–8. Further, we have made an effort to "homogenize" the different data sets by applying area reduction factors in a transparent way, see Section 3.3. For these reasons, the evaluation is qualitative and we focus on the main characteristics and not minor deviations, which we will include a statement about in Section 4.1 Evaluation in the revised manuscript.

---

## Author Comment (AC2) · 13 Mar 2019

**We appreciate very much the comments and suggestions, as well as the time and energy spent in reviewing our manuscript. Below are answers to all items raised.**

General comments: The representation of sub-daily precipitation extremes and their future changes are investigated using a subset of EURO-CORDEX 0.11 climate models. The article gathers an impressive number of datasets and hourly-output from models to assess the limits to the use of convection-parameterised models at sub-daily time-scales in summer, which had never been done before. The authors first provide an evaluation of depth-duration-frequency curves (return-levels) against pre-calculated

[Figure]

country-wide DDF curves. They conclude that convection-parameterised models at 0.11 are not able to represent hourly intense rainfall events: they mostly underestimate 10-year return level precipitation. Their ability is mostly RCM dependent. However, the models show skills in representing 12-hourly return values. The authors show that the 12h return value is increasing with temperature in future climate scenarios, but that the slope depends both on the RCM and the GCM. Although not reliable, hourly intensities increase generally at a larger rate than 12-hourly intensities. This study introduces an interesting methodology and comparison with observations which could be further used in the assessment of future convection-permitting ensembles of models. I find the article scientifically robust, written in a clear manner and worth of publication in NHESS. I mainly have minor comments, which I believe could improve the manuscript.

Specific comments: 1) P6L19-21: Is the 3h separation for values below 3h enough to assume "iid"? You write that this is higher than many studies, but it is lower than Ban et al (2018) (2days) or Chan et al. 2014 (1day). Does using 1 day for all durations significantly impact the results? L21-22 is not comprehensible. Please clarify.

**There have indeed been many different choices made for the time separation between events, and our choice is justified mainly in comparison to the literature presented on Page 6 lines 21–22. However, a clarification is in place here: the separation stated as x-hours is on both ends of the event, so for 1h durations, a period of 7 hours (3h before, the actual event and 3h after) is used to exclude further events. For 6h duration, a period of 18 hours is excluded. This clarification will be included in the revised manuscript. Further, we are interested in evaluating the models to the gathered observational data, and our choice for event separation is therefore mimicking their choices, as presented in Section 2.2. We will include this explanation in Section 3.2 together with the clarification above.**

**We doubt that the results would be significantly altered by using a 24 hour separation. As explained above, we are close to or beyond such a separation for the 6h and 12h durations, so these durations would not be altered. The shorter du-**

**rations might be, but some samples we made on the actual separation between the selected events show that although several events are close to the separation limit (e.g. 3h), the small peak we see close to the limit is not peaking at the limit, but some steps away from it. This indicates that the events are not part of the same peak precipitation period, but rather two peaks, or separate events, in succession.**

**Page 6 lines 21–22 are stating that we are conservative in our choices with our fix time step 1h data and fix duration block rains, compared to studies using event durations defined by connected time periods above a set threshold. This will be clarified in the revised manuscript.**

2) P4 L17: "The analysis is restricted to summer-half years (April–September) to focus on the main convective season in Europe (Berg et al., 2009)." Note that you are missing most of the season of deep convective events in the Mediterranean (Sept.-Dec.): it may be worth producing the French map or Europe-wide map for this season, or extending the season to October. e.g.: Enno, Sugier and Alber (2018) Lightning flash density in Europe on the basis of 10 years of ATDnet data; 25th international lightning detection conference 7th international lightning meteorological conference You could also note that seasonality changes, such as reported by Marelle et al. (2018) are not taken into account in your study.

**That is a very good point. We do not have the resources to redo our analysis for this paper since the calculation are quite time consuming, but will mention this unfortunate cut-off for the Mediterranean climate in the revised manuscript. Thank you for the references which we will also include.**

3) This is a semantic question, but I find the term "cloud burst" in the introduction rather ill-defined, it seems to be defined by its impacts, and to correspond to convective rainfall above 100mm/h? 50mm/h? 12h duration rainfall is probably more like frontal rainfall in most european regions, is this a "cloud burst"? I would use the term "heavy

precipitation event" or extreme precipitation event, which is probably less dependent on the type of precipitation event/the impacts it has.

**We agree, and will change accordingly in the revised manuscript.**

4) P2L34-35: Ban et al. (2018) do not find a stronger scaling for intense events in convection-permitting models compared to convection-parameterised models, to the contrary it is weaker in summer, which is your season of interest.

**Thanks for noticing this, we will revise this sentence.**

5) Figures 2-7 and S1-4: you show continuous fields with a diverging color-bar, this can me a bit misleading, please use a sequential (multi-hue) colorscale. https://journals.ametsoc.org/doi/pdf/10.1175/BAMS-D-13-00155.1 You could start the colorscale above 0 to use all the colour intervals in the figure.

**We will take this into account when revising the figures to make them as intuitive as possible.**

6) It'd be interesting to see the spatial variability of the precipitation enhancement thanks to the additon of a map of future changes (e.g. 10-year return value of 12h-duration) for RCP8.5. In Fig. 8, you are pooling the results in a single figure, on which it is difficult to see individual regions (you could reduce the y limits to 60

**Thanks for the suggestions. We will adjust the vertical limits for increased readability in Fig. 8. And we will consider including a map of the percentage scaling per grid point for rcp8.5, 10-year return value and 12h duration as suggested.**

7) P2L20-21: you could add that convection-permitting models better represent Mediterranean heavy precipitation events (which stand out in your Fig. 4-5) and in some regions still overestimate moderate to intense hourly precipitation (Berthou et al. 2018).

**Thank you for the reference, which we will include as suggested.**

Technical corrections: P2L16: add "in Sweden". P8L9: parameters fits -> parameter fits P10L15: intra-RCM spread -> inter-RCM spread P12L9: the core of the events -> the peak of the events P6L24: de Haans -> de Haan P6L24: Picklands (1975) not referenced

**Thank you. We will adjust accordingly.**

Berthou, S., Kendon, E. J., Chan, S. C., Ban, N., Leutwyler, D., Schär, C., Fosser, G. (2018). Pan-European climate at convection-permitting scale: a model intercomparison study. Clim. Dyn. http://doi.org/10.1007/s00382-018-4114-6

Chan, S. C., Kendon, E. J., Fowler, H. J., Blenkinsop, S., Roberts, N. M. (2014). Projected increases in summer and winter UK sub-daily precipitation extremes from high-resolution regional climate models. Environmental Research Letters, 9(8), 84019.

Marelle, L., Myhre, G., Hodnebrog, Ø., Sillmann, J., Samset Bjørn, H. (n.d.). The changing seasonality of extreme daily precipitation. Geophysical Research Letters, 0(ja). http://doi.org/10.1029/2018GL079567

---

## Author Comment (AC3) · 13 Mar 2019

**We appreciate very much the comments and suggestions, as well as the time and energy spent in reviewing our manuscript. Below are answers to all items raised.**

The manuscript of Berg et al. provides a comparison between regional climate model outputs of precipitation and high-resolution observational datasets in Sweden, Germany, Austria, Netherland and France. Overall, the manuscript is well written, the objectives are clear and the results support the goals of the study. Yet, I am puzzled with this submission since to my opinion it does not bring new results. Indeed, the conclusion can be found in the introduction, page 2, line 13-19: "However, RCMs and GCMs

[Figure]

have shown severe problems with their sub-grid scale parametrisations of convective processes, which affect their ability to reproduce, e.g., the diurnal cycle of rainfall intensity (Trenberth et al., 2003; Fosser et al., 2015; Prein et al., 2015), the peak storm intensities (Kendon et al., 2014), and extreme hourly intensities (Hanel and Buishand, 2010). It is therefore questionable to which extent suchRCMs are capable of describing cloudbursts in present as well as in future climate" Indeed, it is well known that the current generation of CORDEX RCMs includes a convective scheme that is not able to reproduce adequately the small-scale high-intensity rainfall events. Beranová et al. (2018) evaluated the hourly outputs of RCMs and projections for short duration's rainfall have been provided by KyselÃÂą et al. (2012), among others. This is the reason why regional climate models that explicitly reproduce convection are being developed, there is a huge amount of literature presenting this new generation of climate models, see for instance Coppola et al. 2018 or Berthoux et al. 2018 (I believe both should be cited in the text). However, I agree as stated by the authors page 3, line 1 that the convection-permitting simulations are still not widely available, unlike EuroCordex runs. Yet, when reading the manuscript it seems that these convection-permitting simulations are still not available for research purpose, when several studies have already been produced with these types of model (see Berthoux et al. 2018, Reszler et al. 2018). It can be somewhat misleading to the reader not familiar with climate models.

**We agree on many of the raised points; there are earlier studies that have addressed deficiencies in the sub-daily precipitation of parameterized models for different regions and statistics, and the reviewer provides references to additional studies that will be included in the revised version. What separates the current study from earlier ones is (i) the novel method of evaluation with national data sets of extreme precipitation statistics, (ii) the spatial analysis for part of the data sets in (i), (iii) a larger set of state-of-the-art paramaterized RCMs with intercomparison of the models, (iv) identification of which time-scales (duration) that are better captured with these models and can with high confidence be used for climate change assessments, and (v) analysis of the sensitivity to a changing**

**climate. Points (i), (ii), (iv) and (v) are readily applicable to future evaluations of convection permitting simulations.**

**We believe that these points are already well described in the current manuscript, as also noted by other reviewers. We will clarify the point about the convection permitting simulations being available in the research community, and include the suggested references.**

Specific comments: Since the study focuses on the summer season, the title should say it. In various regions such as south France, the maximum intensity events are occurring in the autumn, not during summer.

**That is a very good point. We do not have the resources to redo our analysis for this paper since the calculation are quite time consuming, but will mention this unfortunate cut-off for the Mediterranean climate in the revised manuscript. Thank you for the references which we will also include.**

Page 4, line 6: Rajczak and Shär 2017 analyzed daily model outputs

**The line does states that: "Rajczak and Shär (2017) analysed heavy and extreme daily precipitation intensity..."**

Page 3 section 2.1: it should be clearly stated here that the 9 simulations all include a parametrized convection scheme.

**We will add this in the manuscript to make this very clear.**

Page 13, lines 9: it is not clear which threshold is used in the GP model for future time periods. As explained page 7, lines 7-14, a precipitation threshold is defined for each grid point to have 3 events on average per year. Which value is used for the future time period? the threshold value yielding 3 events per year in present climate ? The authors should provide, at least in the text, the ranges of threshold values obtained for the different grid points/regions.

**We treat the different time periods separately, so the threshold is unique for each time-slice and therefore also different for historical and future projections. At 1h duration the thresholds range from about 1 to 30 mm/h across land regions of Europe and across all models for the historical period, with domain median values of about 3 to 7 mm/h. The largest changes are towards the end of the century in RCP8.5 where the domain median values increase by between 13 and almost 50% across the models. At 12h duration, the thresholds range from about 0.5 to 10 mm/h and medians between 1.4 to 1.8 mm/h. The change under RCP8.5 range from 14 to almost 50%, similar to the 1h duration. We will include this information in a comprehensible way in the revised text.**

Page 15, line 13-15: it is very good that the authors talk about data availability in the discussion. It should be stressed also that the different data sets they used are probably not homogeneous at all: some rely on observed precipitation, some rely on a mixture of observed precipitation and simulations from a climate model (Germany) and some rely on a weather generator (France). Further work should try to homogenize these data sets prior to the evaluation of climate models, or the discrepancies between data set could induce an artificial bias in the evaluations. Due to different sources of data, is it very likely that the spatial patterns of the different datasets cannot be compared in a robust way.

**We agree, and already touch upon this in the discussion in Section 5, but will explicitly mention the issue of homogeneity between methodologies to allow more direct comparisons.**

References: Berthou, S., Kendon, E. J., Chan, S. C., Ban, N., Leutwyler, D., Schär, C., and Fosser, G.: Pan-European climate at convection-permitting scale: a model intercomparison study, Clim. Dynam., https://doi.org/10.1007/s00382-018-4114-6, in press, 2018

Beranová R., Kyselý J., Hanel M., 2018: Characteristics of sub-daily precipitation

extremes in observed data and regional climate model simulations. Theoretical and Applied Climatology, 132, 515-527

Coppola, E., Sobolowski, S., Pichelli, E., Raffaele F., Ahrens, B., Anders, I., Ban, N., Bastin, S., Belda, M., Belusic, et al., 2018. A first-of-its-kind multi-model convection permitting ensemble for investigating convective phenomena over Europe and the Mediterranean. Climate Dynamics https://link.springer.com/article/10.1007/s00382-018-4521-8

KyselÃÂą J., Beguería S., Beranová R., Gaál L., López-Moreno J.I., 2012: Different patterns of climate change scenarios for short-term and multi-day precipitation extremes in the Mediterranean. Global and Planetary Change, 98-99, 63-72

Reszler, C., Switanek, M. B., and Truhetz, H.: Convection-permitting regional climate simulations for representing floods in small- and medium-sized catchments in the Eastern Alps, Nat. Hazards Earth Syst. Sci., 18, 2653-2674, https://doi.org/10.5194/nhess-18-2653-2018, 2018.